# Retinal Function in Long-Term Type 1 Diabetes without Retinopathy: Insights from Pattern Electroretinogram and Pattern Visual Evoked Potentials Assessments

**DOI:** 10.3390/diagnostics14050492

**Published:** 2024-02-25

**Authors:** Marta Arias-Alvarez, Maria Sopeña-Pinilla, Guisela Fernandez-Espinosa, Elvira Orduna-Hospital, Ines Vicente-Garza, Anna Bonet-Rodriguez, Javier Acha-Perez, Diego Rodriguez-Mena, Isabel Pinilla

**Affiliations:** 1Department of Neurophysiology, Lozano Blesa University Hospital, 50009 Zaragoza, Spain; martariasalvarez7@gmail.com (M.A.-A.); inesvicente3@gmail.com (I.V.-G.); annabonet@msn.com (A.B.-R.); drodriguezm@salud.aragon.es (D.R.-M.); 2Aragon Institute for Health Research (IIS Aragon), 50009 Zaragoza, Spain; mariasopenapinilla@gmail.com (M.S.-P.); guisela.fernandez3@gmail.com (G.F.-E.); eordunahospital@unizar.es (E.O.-H.); j.acha.perez@gmail.com (J.A.-P.); 3Department of Ophthalmology, Miguel Servet University Hospital, 50009 Zaragoza, Spain; 4Department of Applied Physics, University of Zaragoza, 50009 Zaragoza, Spain; 5Department of Endocrinology, Miguel Servet University Hospital, 50009 Zaragoza, Spain; 6Department of Surgery, University of Zaragoza, 50009 Zaragoza, Spain; 7Department of Ophthalmology, Lozano Blesa University Hospital, 50009 Zaragoza, Spain

**Keywords:** type 1 diabetes, diabetic retinopathy, pattern electroretinogram, pattern visual evoked potentials

## Abstract

Background: To evaluate changes in pattern electroretinogram (pERG) and pattern visual evoked potentials (pVEP) in patients with long-lasting type 1 diabetes without diabetic retinopathy (DR). Methods: Prospective study involving 92 eyes divided into two groups. The diabetic group included 46 eyes of 23 patients with type 1 diabetes (T1DM); the control group included 23 age-matched healthy subjects. pERG and pVEP were assessed using the RETI-port/scan21 recording software (version 1021.3.0.0). Results: Mean age was 48 ± 9.77 years for the diabetic group and 51.7 ± 4.75 years for the control group. The mean duration of diabetes was 28.88 ± 8.04 years. The mean HbA1c value was 7.29 ± 0.89%. There were no differences in the age or sex distribution. Regarding the pERG, T1DM patients exhibited a significant decrease in the amplitude of the P50 and N95 waves compared to the control group (*p* = 0.018 and *p* = 0.035, respectively), with no differences in the peak time of each component. pVEP showed no significant changes in either peak time or amplitude of the different components. Conclusions: Long-term T1DM patients without DR showed changes in the amplitude of pERG waves with preserved peak times. We did not observe modifications in pVEP. pERG may serve as a subclinical marker of ganglion cell damage in long-term T1DM patients.

## 1. Introduction

Diabetic retinopathy (DR) is a neurovascular complication of diabetes mellitus (DM) associated with the duration of the disease [1]. It is recognised as the primary cause of visual loss in working-age adults [2]. The pathogenesis of DR involves two processes: a neuropathic and a vascular process; the relationship between these two is not well understood [3,4,5]. Before the onset of DR, subtle anatomical changes can be detected using devices such as optical coherence tomography (OCT) and OCT angiography (OCTA). These alterations include the loss of the ganglion cell complex or microvascular abnormalities [6,7]. In addition to anatomical changes, functional defects have been observed through other examinations such as microperimetry [8,9], contrast sensitivity [10] and colour perception tests [11,12]. Electrophysiological abnormalities are common findings in long-standing type 1 diabetic (T1DM) patients with no signs of DR. 

Visual neurophysiological studies involve various tests which include a full-field electroretinogram (ffERG), multifocal ERG (mfERG), electrooculogram (EOG), pattern ERG (pERG) and visual evocated potentials (VEP). These tests explore structures within the eye and the visual pathway. ffERG provides information about photoreceptors and the changes occurring in their postsynaptic neurons after light stimulation and hyperpolarisation. mfERG offers information about cone function in a light-adapted background. EOG is based on potential changes between the cornea and the back of the eye, providing insights into the status of the retinal pigment epithelium (RPE) [13,14]. 

pERG investigates how retinal cells respond to a phase-reversing patterned stimulus, which can be a grating or checkerboard stimulus. It predominantly reflects ganglion cell activity and has three major components: N35 (a negative corneal wave at 30–35 ms), P50 (a major positive wave that appears around 50 ms) and N95 (a negative peak at 95 ms) [15]. However, the pERG does not solely reflect how the ganglion cells respond. While N95 is linked to ganglion cell function, P50 provides information about the more external retina [16]. 

VEP studies the electroencephalographic activity in the visual cortex and reflects the transmission along the visual pathway [17]. Pattern VEP (pVEP) is the preferred protocol due to its minimal waveform and peak time variability, as well as its excellent reproducibility [17,18]. 

Following foveal stimulation, the occipital electrode records two negative waves and one positive wave after different peak times (N75, P100 and N145 at 75, 100 and 145 ms, respectively). However, the P100 amplitude and peak time are considered the most reliable indicators for detecting clinically significant changes in the visual pathway [18]. The P100 waveform originates in the striate and peristriate occipital cortex, triggered by the activation of the primary visual cortex. The N70 component reflects the activity of the fovea and primary visual cortex, whereas the N145 component reflects the activity of the visual association cortex [17]. 

The inner retinal layers are the most sensible to alterations that occur in T1DM patients. Retinal ganglion cells are the earliest affected and show the highest apoptosis rate, although changes in the outer retina have also been described [19,20]. OCT can identify the loss of ganglion cells and thickness reduction in the retinal nerve fibre layer (RNFL), even before the initial signs of DR become visible [21,22,23,24,25,26,27]. pERG has the capability to assess ganglion cell function, which is considered a limitation of both ffERG and mfERG. The simultaneous use of pVEP and pERG is useful for identifying the location of lesions.

The presence of neurophysiological changes in DM patients without DR has been studied over the years [28]. There is evidence that electrophysiological tests are sensitive and reliable markers of retinal or optic nerve involvement in DM patients even before the onset of DR. However, pERG results in DM patients without DR have yielded controversial findings and it is challenging to find studies involving long-standing T1DM patients.

The aim of our study was to evaluate ganglion cell and optic nerve function using pERG and pVEP in long-standing T1DM patients without DR. 

## 2. Materials and Methods

This investigation adhered to the principles outlined in the Declaration of Helsinki and obtained approval from the Ethics Committee for Clinical Research of Aragon (CEICA PI22/587). Prior to examinations, written informed consent was obtained from all participants.

A prospective observational study was conducted from October 2022 to May 2023, focusing on visual function in 46 eyes from 23 patients with T1DM and 46 eyes from 23 normal age-matched controls. The evaluation took place at the Neurophysiology Department of the Lozano Blesa University Hospital (Zaragoza, Spain), conducted consistently by the same investigator (MAA).

The T1DM patients were under the monitoring of the Endocrinology Unit, maintaining well-controlled glycated haemoglobin (HbA1c), lipid values and arterial blood pressure. All participants underwent comprehensive ophthalmological and neurophysiological examinations, encompassing medical history, best-corrected visual acuity (BCVA), axial length (AL) measured with IOLmaster 500 (Carl Zeiss Company, Jena, Germany), slit-lamp examination, intraocular pressure (IOP) measurement with Goldmann applanation tonometry, fundus examination and wide-field retinography with Clarus 700 (Carl Zeiss Meditec, Dublin, OH, USA), macular thickness using Spectralis OCT (Heidelberg Engineering, Heidelberg, Germany), pERG and pVEP.

The T1DM patients had a minimum disease duration of 20 years, with no vascular changes in the retina. Their BCVA had to be over 20/25 on the Snellen chart and their refractive errors had to be less than 5D of spherical equivalent or 3D of astigmatism. All of them provided informed consent. The control group consisted of healthy subjects matched in age to the T1DM group, all of whom were either healthcare personnel or family members of the patients, with no ocular or systemic diseases. They met the same inclusion criteria, excluding the T1DM diagnosis. Informed consent was obtained from all participants.

Exclusion criteria for both diabetic and control groups included the presence of any sign of DR or other ocular pathologies, IOP exceeding 21 mmHg on Goldmann tonometry, optic nerve pathology or any pallor suggesting diabetic neuropathy, ocular inflammation, and any prior ocular surgeries.

All the neurophysiological assessments were performed with the RETI-port/scan21 recording software (Roland Consult, Brandenburg, Germany). Our protocols followed the guidelines set by the International Society for Clinical Electrophysiology of Vision (ISCEV), which incorporated updates for pERG and pVEP standards in 2013 [15] and 2016 [18], respectively. 

Simultaneous recordings of pVEP and pERG were chosen on separate channels. Prior to each session, subjective refractive errors were corrected with lenses. The recordings were performed without mydriasis. Participants were positioned one meter away from a television screen placed at eye level in a dim and quiet room. 

After cleaning the scalp to maintain impedance below 5 kΩ, gold-cup skin electrodes were placed according to the International 10/20 system. For pVEP recordings, the active electrode was placed on the occipital scalp at Oz, the reference electrode at Fz, and the ground electrode at Cz. Anterior/posterior midline measurements were based on the distance between the nasion and the inion over the vertex.

For pERG recordings, following topical anaesthesia, active sterile DTL electrodes were carefully positioned across the bulbar conjunctiva. A gold-cup skin reference electrode was placed superotemporally or laterally to the orbital rim, with a ground electrode on the forehead. 

The standard pattern stimuli featured a high-contrast black-and-white checkerboard with a 1-degree check size. The amplifier bandpass settings were 1 to 100 Hz. The photopic luminance for the light and dark checks was 96 cd/m^2^, and 2.4 cd/m^2^, respectively, yielding a Michelson contrast of 97%. The reversal rate was 3.0 reversals per second (rps). The mean of the width and height of the stimulus field was 15 degrees. At least 50 sweeps were acquired and averaged for each subject. The acquisition time window was 250 ms. Recordings followed monocular full-field stimulation, with each eye tested separately. Participants were instructed to focus on a central red mark.

Amplitudes (μV) of standard pERG components were measured between peaks and troughs. N35, P50 and N95 were selected manually. P50 was measured from the trough of N35 to the peak of P50, while the amplitude of N95 was measured from the peak of P50 to the trough of N95. Peak times (ms) were measured from the onset of the contrast reversal to the peak of the respective component.

The amplitudes (μV) of standard pVEP components were assessed peak-to-peak. N75, P100 and N135 were selected manually. P100 was measured from the peak of N75 to the peak of P100, while the amplitude of N135 was measured from the peak of P100 to the peak of N135. The peak times (ms) of N75, P100, and N135 components were determined from the onset of contrast reversal to their respective peaks (Figure 1). 

STATISTICS: Data were entered in a Microsoft Office Excel 2011 spreadsheet (Microsoft Corporation, Redmond, WA, USA). Statistical analysis was conducted using the Statistical Package for Social Science (SPSS) version 22.0 (SPSS Inc., IBM Corporation, Sommers, NY, USA). Normal distribution was assessed using the Kolmogorov–Smirnov test. For comparisons, the Mann–Whitney U test was used after verifying that the data did not meet the criterion of a normal distribution. Spearman’s rho test was employed to calculate the bivariate correlation coefficients.

## 3. Results

The T1DM group comprised 46 eyes from 23 patients. The age at diagnosis was 17.96 ± 13.43 (range 2–47), and the average disease duration was 28.88 ± 8.04 years (range 18–47). The mean age was 48 ± 9.77 years (range 28–69), with a sex distribution of 14 females (60.8%) and 9 males (39.13%). Their mean HbA1c value was 7.29 ± 0.89% (range 6.2–10), and blood pressure and lipid levels remained within normal limits. Other metabolic parameters are presented in Table 1. The control group comprised 46 eyes from 23 healthy subjects, with 14 females (60.8%) and 9 males (39.13%); the mean age was 51.7 ± 4.75 years (range 40–59). There were no differences in the age or sex distribution between both groups. A statistically significant decrease in BCVA was observed in the T1DM group (0.03 ± 0.06 LogMAR) compared to the control group (−0.01 ± 0.04 LogMAR) (*p* = 0.001). No differences were found in IOP (15.76 ± 2.14 and 15.75 ± 3.42 in the T1DM and control groups, respectively; *p* = 0.614).

Regarding the pERG, we observed a significant decrease in the amplitude of the P50 (N35-P50) and N95 (P50-N95) waves in the T1DM patients compared to the control group. The N35–P50 amplitude was 4.72 ± 2.47 and 3.62 ± 2.10 µV in the control and T1DM groups, respectively (*p* = 0.018). The P50–N95 amplitude was 9.32 ± 3.45 and 7.90 ± 3.17 µV in the control and T1DM groups, respectively (*p* = 0.035). However, no differences were observed in the peak time of the waves between the two groups (Table 2, Figure 2).

Regarding the correlation study, a negative correlation was found between age and the peak-to-peak P50 (N35-P50) and N95 (P50-N95) components (r = −0.399 *p* = 0.006; r = −0.613 *p* < 0.001, respectively). Additionally, there was a positive correlation with the N35 peak time (r = 0.490 *p* = 0.001) and P50 peak time (r = 0.390 *p* = 0.007). The level of HbA1c was negatively correlated with the peak-to-peak P50 (N35-P50) component (r = −0.482 *p* = 0.013). The duration of the disease showed no correlation with the amplitude or the peak time of the waves. 

When studying the pVEP, we did not observe significant changes in the peak time or amplitude of the different components, although the peak time values were slightly higher in the diabetic group (Table 3). 

A negative correlation was found between age and the peak-to-peak P100 (N75-P100) and N135 (P100-N135) components (r = −0.628, *p* < 0.001; r = −0.62,9 *p* < 0.001, respectively). Additionally, there was a positive correlation with all peak time components (P50: r = 0.390, *p* = 0.007; N75: r = 0.330, *p* = 0.025; P100: r = 0.735, *p* < 0.001; N135: r = 0.363, *p* = 0.013). Furthermore, HbA1c showed a positive correlation with the P100 peak time (r = 0.473, *p* = 0.015) and N135 peak time (r = 0.435, *p* = 0.026). For the pERG, HbA1c showed a negative correlation with the N35–P50 amplitude (r = −0.482; *p* = 0.013). The duration of the disease showed no correlation with any of the wave’s peak times. 

## 4. Discussion

In our study involving long-standing T1DM patients without evident signs of DR, we observed pERG modifications including a reduction in both P50 and N95 amplitude, despite not finding alterations in their peak times. Unlike other electrophysiological tests such as ffERG or mfERG, ganglion cell function can be assessed through pERG and the photopic negative response (PhNR). pERG findings in T1DM patients have not been extensively studied in comparison to other tests; understanding how ganglion cells function can be crucial, as the loss of ganglion cells has been described as an initial change in diabetic neurodegeneration. pERG has also been used as a glaucoma detection method in DM patients, revealing modifications in its amplitudes [29]. To avoid potential confounding factors, we rigorously excluded all patients exhibiting any indications of glaucoma, such as alterations in IOP levels or any changes in the optic nerve head. 

Previous research has linked pERG changes to the presence of DR. Mermeklieva et al. [30] studied 84 type 2 diabetes mellitus (T2DM) patients with different stages of DR. They observed changes in all pERG components, with severity correlating to the degree of disease progression. DM patients without DR exhibited a reduced amplitude in the P50–N95 component at 15° and 30° and N35–P50 component at 15°. Additionally, these patients showed an increased peak time of the N35. In advanced DR stages, longer peak times and diminished amplitudes were observed across all components. 

Other authors have also documented changes in pERG among DM patients without DR [31,32]. Lecleire-Collet et al. assessed retinal function in 28 patients, including both T1DM and T2DM individuals without DR, revealing changes in peak times and amplitudes among DM subjects. They noted delays in both P50 and N75, along with smaller amplitudes [33]. Lasta et al. [34] studied 50 T1DM patients without DR, with a shorter duration of the disease (9.8 ± 3.4) and similar HbA1c levels (7.5 ± 1.3). They did not find changes in peak time or wave amplitude. The shorter duration of the disease in their sample could explain the differences in the results. In a study by Deak et al., two groups of T1DM patients without DR were compared based on the presence or absence of neuropathy [35]. Patients with polyneuropathy exhibited alterations in VEP in all cases, but only 50% showed changes in pERG. Of the patients without polyneuropathy, 95% had abnormal pERG results. Their common findings included a delayed P50 peak time and diminished P50 and N95, suggesting that both tests are highly sensitive tools for evaluating neuronal damage even before the onset of DR. Park et al. found changes in pERG only in patients with vascular lesions in a study involving 45 T2DM subjects [36]. Tabl also reported a delay in N95 in 30 eyes of T2DM patients [37]. Caputo et al. [38] observed a reduction in pERG amplitude among T1DM patients with no DR compared to the control group, as well as between patients with minimal changes and those without lesions. Similar findings were reported by Falsini et al., particularly in relation to the age of disease onset [39]. Parisi and Uccioli identified these abnormalities in T1DM patients with a disease duration shorter than 6 months, making it the first detectable electrophysiological alteration [40]. Additionally, after one year of the disease, they observed changes in focal ERG and VEP following photostress, while ffERG and oscillatory potentials remained normal. Their findings suggested an early dysfunction of the innermost retinal layers without impairment of preganglionic elements [40]. pERG is capable of detecting early macular dysfunction in DM patients, and these alterations intensify with the progression of DR [28,38].

In our study, we did not identify any correlation between the duration of the disease and the amplitude or the peak time of the waves. Different results could arise from variations in the stimuli parameters employed for pattern stimulation, including factors such as check size and contrast level. We observed a negative correlation between the level of HbA1c and the peak-to-peak P50 (N35–P50) component. This specific correlation has not been reported by other authors [40]. 

In numerous studies, a decrease in amplitude and an increase in the peak time of VEP waves have been observed in DM patients with varying degrees of DR. Peak time changes are the most significant outcome in VEP, with many authors reporting delays in DM patients even before the onset of DR [17,40,41,42,43,44,45]. Gupta et al. [43] demonstrated an increase in P100 peak time associated with glycaemic control in DM patients without DR, along with a decrease in the N75–P100 amplitude. Balta et al., studying 58 young-adult T2DM patients without DR, found an increase in both P100 and N135 peak times for the 60 min and 120 min check sizes, respectively. They also identified correlations between diabetes duration and P100 peak time in the 15 and 7 min check sizes, and the N135 peak time in the 15 min check size [46]. Delayed peak times observed in DM patients are believed to indicate delayed transmission along the optic nerve pathway, consistent with findings from studies using pERG [47] and retinocortical times [48]. Reduced VEP amplitudes in certain DM patients imply dysfunction of functional fibres in the visual pathway, which occur before the identification of anatomical abnormalities [17]. Martinelli et al. suggested that changes in VEP in T1DM patients are related to structural involvement of the visual pathway, with no relationship with acute short-term hyperglycaemia [49]. 

These findings suggest that the VEP peak time could serve as a more sensitive indicator of alterations, retinal ganglion cell damage and optic nerve pathways during the initial phases of DM. Heravian et al. suggested that pVEP could be used for detecting prediabetic retinopathy, with an increased peak time indicating retinal ganglion cell damage [44]. In their study, they found a longer P100 peak time in DM patients without differences in the N75 latency. No correlations were observed between peak time and disease progression or glycaemic levels. The patients had a mean age of 54.8 ± 8.2 years, and their study did not differentiate between T1DM and T2DM subjects.

Although the disease duration in our sample was extended, we did not find an increase in the VEP peak time. However, we observed slightly increased peak times across all VEP components (N75, P100 and N135) without statistically significant differences. Abnormalities in the VEP peak times are deemed more important than changes in amplitude [46]. Various factors contribute to the differences in our results. Firstly, our diabetic patients were T1DM subjects, potentially differing from T2DM patients. Secondly, patients with changes in the optic nerve head were excluded from our study, ruling out signs of optic nerve neuropathy related to DM. Additionally, our patients maintained a moderate-to-good glycaemic control.

When examining the peak times in our study, we observed a negative correlation with age and a positive correlation of both P100 and N135 peak times with HbA1c levels. We did not find correlations with the duration of the disease. Previous reports presented different findings [17]. Lee et al. observed a correlation between the P100 peak time and HbA1c levels, but this association was observed only in T2DM patients, not in those with T1DM [50]. The variability in the reported outcomes suggests that the peak time might serve as an indicator of HbA1c levels, although its connection with long-term glycaemic control or disease duration remains unclear [17].

DR is acknowledged as a neurovascular disease [1]. The central nervous system is organised into neurovascular units, which consist of neurons, glial cells and vascular cells. The neurovascular units are often compromised in neurodegenerative diseases such as stroke, Alzheimer’s or Parkinson’s disease. Neurovascular coupling has been shown to deteriorate in the initial stages of DM. Changes in retinal vessel dilation induced by flicker stimulation are known to occur in the early stages of DM, even in the absence of DR [51,52]. Neurophysiological findings from other tests such as ffERG or mfERG are commonly present in these patients before the emergence of vascular signs. These findings include changes in the oscillatory potentials, recognised as a marker of disease progression [53], as well as a diminished response to a higher-frequency flicker or dark-adapted flicker due to a reduced response of retinal arteries to this flicker stimulation [54,55,56]. Our findings support evidence of ganglion cell dysfunction, which, together with the anatomical changes observed in the inner retina, may be related to modifications in neurovascular coupling. Our findings in the studied neurophysiological test could be related to other retinal anatomical exams, including OCT. Retinal images show inner retinal modifications, including loss of the ganglion cell layer and thinning of the retinal nerve fibre layer. These observed anatomical changes could explain the neurovisual functional deficits observed using different tests. Retinal structure and function should be studied together to recognise the modifications that appear prior to the onset of signs of DR. All these changes that occur prior to the onset of DR, including alterations in retinal layer thicknesses, neurophysiological modifications or variations in the retinal capillary layers studied with non-invasive techniques such as OCTA, could offer predictive data regarding both vascularity and neuronal status in T1DM.

A limitation of our study is the relatively small sample size. More patients should be tested and followed to assess the progression of our findings.

In our current investigation, pERG emerged as a sensitive indicator of retinal implications in DM, even in an early stage when patients did not show evident signs of retinal involvement during fundus examination. VEP may be more closely linked to peripheral nerve damage and could share a common pathogenesis [17,57].

We believe that the combination of pERG and pVEP for ophthalmologic screening in diabetes could be useful, reflecting the compromise of ganglion cells and providing information about the visual pathway as a marker of neurodegeneration or neuropathy. They inform about the level of macular impairment or neuropathy that these patients are experiencing in a shorter time than ffERG or mfERG, and without the need for mydriasis and/or dark adaptation to obtain accurate results.

## 5. Conclusions

In conclusion, long-standing T1DM patients showed changes in the amplitude of pERG waves, with preserved peak times. We were not able to find any modification in the pVEP. pERG could be a marker of ganglion cell damage in long-term T1DM patients.

## Figures and Tables

**Figure 1 diagnostics-14-00492-f001:**
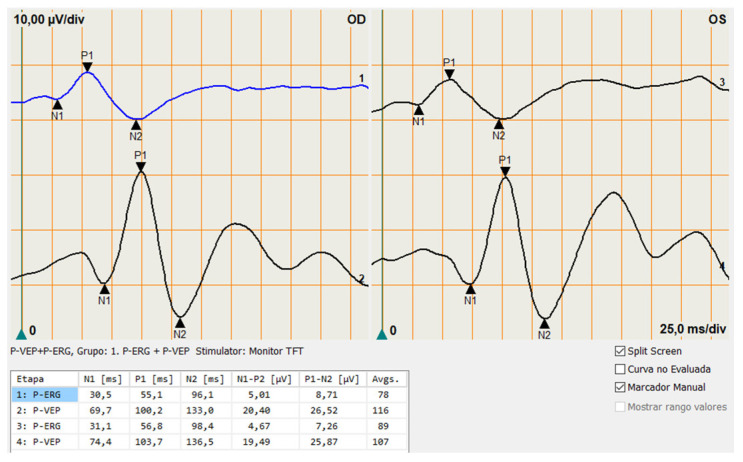
Example of a pERG and a pVEP of a healthy subject from the control group. Both eyes are presented. A series of negative and positive deflections can be appreciated in both registers. The pERG waves are shown on the top register as N35, P50 and N95. The pVEP waves (bottom register) appear as N75, P100 and N145.

**Figure 2 diagnostics-14-00492-f002:**
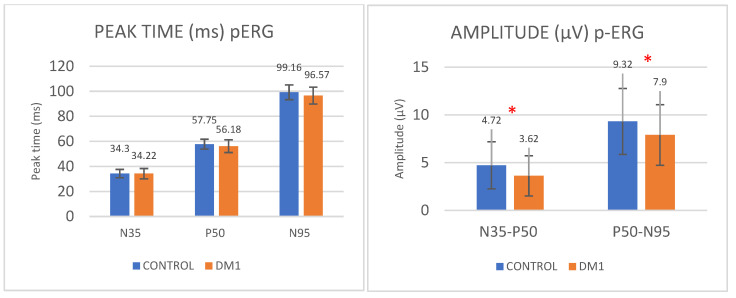
pERG peak time (ms) and amplitude (µV) values obtained with the RETI-port/scan21 recording system in the T1DM (presented in orange) and control groups (presented in blue). Values are expressed as the means ± standard deviation. Abbreviations: pERG, pattern electroretinogram; T1DM, type 1 diabetes mellitus. Significant differences are shown as *.

**Table 1 diagnostics-14-00492-t001:** Mean and standard deviation (SD) of the metabolic parameter of the T1DM group. Abbreviations: HbA1c, glycosylated haemoglobin; HDL, high-density lipoprotein; LDL, low-density lipoprotein; SD, standard deviation.

Type 1 Diabetes Group	Mean ± SD
HbA1c (%)	7.29 ± 0.89
Glycaemia (mg/dL)	149.00 ± 66.13
Total cholesterol (mg/dL)	190.61 ± 33.15
HDL cholesterol (mg/dL)	62.43 ± 12.38
LDL cholesterol (mg/dL)	114.30 ± 27.78
Urea (mg/dL)	34.35 ± 8.62
Creatinine (mg/dL)	0.78 ± 0.10
Albumin/creatinine ratio (mg/g Cr)	7.13 ± 10.19

**Table 2 diagnostics-14-00492-t002:** pERG peak time (msec) and amplitude (µV) values obtained with the RETI-port/scan21 recording system in the T1DM and control groups. Values are presented as the mean ± standard deviation (SD). Differences were considered statistically significant when *p* < 0.05. Statistically significant differences are presented in red and bold. Abbreviations: T1DM, type 1 diabetes mellitus.

	Control Group *n* = 46	T1DM Group *n* = 46	
	Mean ± SD	Mean ± SD	*p*
N35 (ms)	34.30 ± 3.30	34.22 ± 4.10	0.693
P50 (ms)	57.75 ± 3.93	56.18 ± 5.07	0.105
N95 (ms)	99.16 ± 5.90	96.57 ± 6.78	0.93
N35–P50 (µV)	4.72 ± 2.47	3.62 ± 2.10	**0.018**
P50–N95 (µV)	9.32 ± 3.45	7.90 ± 3.17	**0.035**

**Table 3 diagnostics-14-00492-t003:** pVEP peak time (ms) and amplitude (µV) values obtained with the RETI-port/scan21 recording system in the T1DM and control groups. Values are presented as the mean ± standard deviation (SD). Differences were considered statistically significant when *p* < 0.05. Abbreviations: T1DM, type 1 diabetes mellitus.

	Control Group n = 46	T1DM Group n = 46	
	Mean ± SD	Mean ± SD	*p*
N75 (ms)	73.59 ± 7.30	74.36 ± 6.98	0.584
P100 (ms)	108.09 ± 4.70	109.37 ± 7.13	0.56
N135 (ms)	143.68 ± 9.07	144.03 ± 10.42	0.935
N75–P100 (µV)	12.46 ± 4.75	12.23 ± 5.17	0.803
P100–N135 (µV)	12.92 ± 5.03	12.91 ± 6.31	0.87

## Data Availability

The datasets generated and analysed during the current study are not publicly available to protect study participant privacy, but are available from the corresponding author on reasonable request.

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
