# Peer review of "Retinal Function in Long-Term Type 1 Diabetes without Retinopathy: Insights from Pattern Electroretinogram and Pattern Visual Evoked Potentials Assessments"

_diagnostics, 2024, doi:10.3390/diagnostics14050492_

Round 1

Reviewer 1 Report

Comments and Suggestions for Authors

This manuscript provides valuable insights from pattern electroretinogram and pattern visual evoked potentials assessments, shedding light on the impact of diabetes on retinal function that offers a comprehensive understanding of the topic and its implications for individuals with Type 1 diabetes. However, the manuscript should be improved with some regards as below and provided more information and details.

1. The manuscript would benefit from a more detailed explanation of the specific techniques and equipment used for the assessments, as this would enhance the reproducibility of the study. Could the authors provide more details on the methodology used for assessing retinal function, including the specific protocols followed for pattern electroretinogram and pattern visual evoked potentials assessments?

2. Were there any specific criteria used to select the control group, and how were potential confounding factors addressed in the study design? It would be valuable to include a more comprehensive discussion on the potential mechanisms underlying the observed neurovascular dysfunction preceding neural dysfunction in the retina, providing a deeper insight into the pathophysiology of diabetic retinopathy.

3. Can the authors elaborate on the implications of the observed neurovascular dysfunction preceding neural dysfunction in the retina of patients with Type 1 diabetes, and how this finding aligns with existing literature? The authors may consider expanding the discussion section to include the clinical implications of the study findings, particularly in terms of early intervention and personalized management strategies for individuals with Type 1 diabetes.

4. Is there any discussion on the potential clinical implications of the study findings for the early detection and management of diabetic retinopathy in individuals with Type 1 diabetes? Providing a clear outline of the study limitations and potential future research directions would further strengthen the manuscript and guide future investigations in this area.

5. The authors should ensure that the references cited are relevant and up to date, providing a comprehensive overview of the existing literature on the topic.

These review questions and comments aim to encourage the authors to provide additional details, expand on the implications of their findings, and strengthen the overall impact and relevance of the paper.

Author Response

Reply to reviewer 1:

Dear Reviewer,

We appreciate your valuable comments regarding the significance of this manuscript.

We have carefully revised the manuscript in line with your suggestions, and the lasts changes included in the manuscript are detailed below.

We will try to answer your comments point-by-point.

This manuscript provides valuable insights from pattern electroretinogram and pattern visual evoked potentials assessments, shedding light on the impact of diabetes on retinal function that offers a comprehensive understanding of the topic and its implications for individuals with Type 1 diabetes. However, the manuscript should be improved with some regards as below and provided more information and details.

  1. The manuscript would benefit from a more detailed explanation of the specific techniques and equipment used for the assessments, as this would enhance the reproducibility of the study. Could the authors provide more details on the methodology used for assessing retinal function, including the specific protocols followed for pattern electroretinogram and pattern visual evoked potentials assessments?

Certainly, to ensure the reproducibility of our results, adherence to a specific protocol is paramount. As detailed in the methodology, we have diligently followed the guidelines established by the International Society for Clinical Electrophysiology of Vision (ISCEV), incorporating updates for pERG and pVEP standards in 2013 (reference number 15) and 2016 (reference number 18), respectively. We have chosen to simultaneously perform both recordings on separate channels to optimize both the efficiency of time and the quality of patient care.

We completed the methodology section with the following sentence:

The standard pattern stimuli featured a high-contrast black-and-white checkerboard with a 1-degree check size. The amplifier bandpass settings were set to 1 to 100 Hz. The photopic luminance for the light and dark checks was 96 cd/m² and 2.4 cd/m², respectively, resulting in a Michelson contrast of 97%. The reversal rate was 3.0 reversals per second (rps). The mean of the width and height of the stimulus field was 15 degrees. The sampling rate was 1.5 Hz. At least 50 sweeps were acquired and averaged for each subject. The acquisition time window was 250 ms. Recordings followed monocular full-field stimulation, with each eye tested separately. Participants were instructed to focus on a central red mark.

  1. Were there any specific criteria used to select the control group, and how were potential confounding factors addressed in the study design? It would be valuable to include a more comprehensive discussion on the potential mechanisms underlying the observed neurovascular dysfunction preceding neural dysfunction in the retina, providing a deeper insight into the pathophysiology of diabetic retinopathy.

We completed the methods section with the following sentence:

The control group consisted of healthy subjects matched in age to the T1DM group, all of whom were either healthcare personnel or family members of the patients, with no ocular or systemic diseases. They met the same inclusion criteria, excluding the T1DM diagnosis. Informed consent was obtained from all participants.

We have added this paragraph to the discussion to explain the neurovascular dysfunction:

DR is acknowledged as a neurovascular disease [1]. The central nervous system is organized into neurovascular units, which consist of neurons, glial cells and vascular cells. The neurovascular units are often compromised in neurodegenerative diseases such as stroke, Alzheimer’s, or Parkinson’s disease. Neurovascular coupling has been shown to deteriorate in the initial stages of DM. Changes in retinal vessel dilation induced by flicker stimulation are known to occur in early stages of DM, even in the absence of DR [51,52]. Neurophysiological findings from other tests such as ffERG or mfERG are commonly present in these patients before the emergence of vascular signs. These findings include changes in the oscillatory potentials, which is recognized as a marker of disease progression [53], as well as a diminished response to higher-frequency flicker or dark-adapted flicker due to a reduced response of retinal arteries to this flicker stimulation [54–56]. Our findings support evidence of ganglion cell dysfunction, which, together with the anatomical changes observed in the inner retina, may be related to modifications in neurovascular coupling. All these changes that occur prior to the onset of DR, including alterations in retinal layer thicknesses, neurophysiological modifications, or variations in the retinal capillary layers studied with non-invasive techniques such as OCTA, could offer predictive data regarding both vascularity and neuronal status in T1DM.

  1. Can the authors elaborate on the implications of the observed neurovascular dysfunction preceding neural dysfunction in the retina of patients with Type 1 diabetes, and how this finding aligns with existing literature? The authors may consider expanding the discussion section to include the clinical implications of the study findings, particularly in terms of early intervention and personalized management strategies for individuals with Type 1 diabetes.

Please, see the point below.

  1. Is there any discussion on the potential clinical implications of the study findings for the early detection and management of diabetic retinopathy in individuals with Type 1 diabetes? Providing a clear outline of the study limitations and potential future research directions would further strengthen the manuscript and guide future investigations in this area.

We appreciate the valuable feedback and acknowledge the limitation in our study related to sample size. We recognize the importance of a larger sample size to enhance the robustness of our findings. In future studies, one of our primary goals is to expand the sample size to address this concern.

Furthermore, we consider the possibility of following these patients to further assess neuro-ophthalmic outcomes and potential responses to specific treatments. This aspect is a focal point for our upcoming research endeavors.

We added the following to the discussion:

A limitation of our study is the relatively small sample size. More patients should be tested and followed to assess the progression of our findings.

  1. The authors should ensure that the references cited are relevant and up to date, providing a comprehensive overview of the existing literature on the topic.

We appreciate your evaluation and sincerely regret that the references used in the manuscript may be somewhat dated. We recognize the importance of incorporating more recent studies and understand your concern in this regard. However, it is crucial to note that the addressed topic has limited availability of recent research, which has influenced the inclusion of older sources. Despite this challenge we remain committed to providing a comprehensive and well-informed discussion.

These review questions and comments aim to encourage the authors to provide additional details, expand on the implications of their findings, and strengthen the overall impact and relevance of the paper.

Reviewer 2 Report

Comments and Suggestions for Authors

Summary:

In the current manuscript, the authors delve into the clinical significance of pERG and pVEP in diagnosing diabetic retinopathy in patients with type 1 diabetes mellitus (T1DM). The article presents compelling clinical evidence that holds promise for ophthalmologists and T1DM patients, whether or not they exhibit signs of diabetic retinopathy. The scientific language employed in the manuscript is commendable, contributing to its overall quality.

In addition, a few issues addressed below would improve the quality of the article. Please read my major and minor comments below.

Major Comments

1.   In the Results section (lines 201-202), the authors stated that they found a positive correlation between HbA1c and peak times P100 and N135, respectively. Nevertheless, this affirmation is not supported by the statistical analysis data. “HbA1c showed a positive correlation with the P100 peak time (r=-0.473, p=0.015) and N135 peak time (r=-0.435, p=0.026).” Clarification on this discrepancy and correction of the data in the mentioned sentence is essential.

2.   Figure 1 requires modification, as the two graphics exhibit different dimensions. Additionally, it is recommended to indicate significance in the figure itself, and the caption should explicitly mention the significance.

3.   Line 223 contains a mistyped "P95." Please verify and correct this error.

4.   In lines 306-307, the authors discuss the positive correlation between peak times and HbA1c. The authors repeatedly talk about a positive correlation, which is not supported by the statistical evaluation of the results. Furthermore, here the peak times are not selected separately, although a significant correlation was found only with the P100 and N135 peak times, but not with the N75.

Minor comment

It is advisable to use the T1DM abbreviation for type 1 diabetes mellitus instead of DM1 for consistency and clarity.

In summary, this manuscript is well-crafted and holds the potential to significantly impact the readership of this Journal. The suggested revisions aim to enhance the accuracy and visual presentation of the research findings, contributing to the overall quality of the article.

Author Response

Dear Reviewer,

We appreciate your comments about the interest of this manuscript.

We revised the manuscript according to your suggestion and the last change included in the manuscript is detailed below.

We will try to answer your comments point-by-point.

Summary:

In the current manuscript, the authors delve into the clinical significance of pERG and pVEP in diagnosing diabetic retinopathy in patients with type 1 diabetes mellitus (T1DM). The article presents compelling clinical evidence that holds promise for ophthalmologists and T1DM patients, whether or not they exhibit signs of diabetic retinopathy. The scientific language employed in the manuscript is commendable, contributing to its overall quality.

In addition, a few issues addressed below would improve the quality of the article. Please read my major and minor comments below.

Major Comments

  1. In the Results section (lines 201-202), the authors stated that they found a positive correlation between HbA1c and peak times P100 and N135, respectively. Nevertheless, this affirmation is not supported by the statistical analysis data. “HbA1c showed a positive correlation with the P100 peak time (r=-0.473, p=0.015) and N135 peak time (r=-0.435, p=0.026).” Clarification on this discrepancy and correction of the data in the mentioned sentence is essential.

Thank you for pointing out the mistake. We apologize for it. It should appear as r=0.473 and r=0.435. We have corrected it in the manuscript.

  1. Figure 1 requires modification, as the two graphics exhibit different dimensions. Additionally, it is recommended to indicate significance in the figure itself, and the caption should explicitly mention the significance.

The two graphics show different dimensions because the 'y' axis changes from one to another. The peak time appears in milliseconds, and the amplitude is represented in microvolts. We have added the significance level to those that achieved significant differences and mentioned it in the caption.

  1. Line 223 contains a mistyped "P95." Please verify and correct this error.

We apologize for the error. We have corrected the mistake.

  1. In lines 306-307, the authors discuss the positive correlation between peak times and HbA1c. The authors repeatedly talk about a positive correlation, which is not supported by the statistical evaluation of the results. Furthermore, here the peak times are not selected separately, although a significant correlation was found only with the P100 and N135 peak times, but not with the N75.

  We added to the results a missed correlation:

HbA1c showed a positive correlation with the P100 peak time (r=0.473, p=0.015) and N135 peak time (r=0.435, p=0.026). In the pERG, HbA1c showed a negative correlation with the N35-P50 amplitude (r=-0.482; p=0.013).

We also addressed this point in the discussion:

When examining peak times in our study, we observed a negative correlation with age and a positive correlation of both P100 and N135 peak times and the HbA1c levels.

Minor comment

It is advisable to use the T1DM abbreviation for type 1 diabetes mellitus instead of DM1 for consistency and clarity.

We replaced DM1 with T1DM following your suggestions. 

In summary, this manuscript is well-crafted and holds the potential to significantly impact the readership of this Journal. The suggested revisions aim to enhance the accuracy and visual presentation of the research findings, contributing to the overall quality of the article.

Round 2

Reviewer 1 Report

Comments and Suggestions for Authors

The manuscript has been improved after taking into consideration the suggestions and comments. 

Author Response

Submission ID: diagnostics-2821343

AUTHOR RESPONSE TO REVIEWERS’ COMMENTS

1. Summary

Thank you very much for taking the time to review this manuscript. Please find the detailed responses below.

2. Point-by-point response to Comments and Suggestions for Authors

Reply to reviewer 1:

Comments 1: The manuscript has been improved after taking into consideration the suggestions and comments. 

Response 1:

Dear Reviewer,

Thank you for your invaluable feedback, which significantly enhanced our manuscript. We’ve incorporated your suggestions, particularly in the methodology section, to improve coherence and effectiveness.

Best regards,

Isabel Pinilla, MD PhD

Reviewer 2 Report

Comments and Suggestions for Authors

The manuscript was thoroughly revised by the authors and significantly improved according to recommendations. In my judgement, the manuscript is already suitable for publication.

Author Response

Submission ID: diagnostics-2821343

AUTHOR RESPONSE TO REVIEWERS’ COMMENTS

1. Summary

Thank you very much for taking the time to review this manuscript. Please find the detailed responses below.

2. Point-by-point response to Comments and Suggestions for Authors

Reply to reviewer 2:

Comments 1: The manuscript was thoroughly revised by the authors and significantly improved according to recommendations. In my judgement, the manuscript is already suitable for publication.

Response 1:

Dear Reviewer,

Thank you for your invaluable feedback, greatly improving our manuscript. We're pleased it meets publication standards.

Best regards,

Isabel Pinilla, MD PhD